# Development of SCAR Markers for Genetic Authentication of *Metarhizium acridum*

**DOI:** 10.3390/jof10040269

**Published:** 2024-04-04

**Authors:** Conchita Toriello, Esperanza Duarte-Escalante, María Guadalupe Frías-De-León, Carolina Brunner-Mendoza, Hortensia Navarro-Barranco, María del Rocío Reyes-Montes

**Affiliations:** 1Departamento de Microbiología y Parasitología, Facultad de Medicina, Universidad Nacional Autónoma de México, Mexico City 04510, Mexico; toriello@unam.mx (C.T.); dupe@unam.mx (E.D.-E.); brunner.carolina@facmed.unam.mx (C.B.-M.); horte56@yahoo.com.mx (H.N.-B.); 2Hospital Regional de Alta Especialidad de Ixtapaluca, Carretera Federal México-Puebla Km. 34.5, Pueblo de Zoquiapan, Ixtapaluca 56530, Mexico; magpefrias@gmail.com

**Keywords:** *Metarhizium acridum*, SCAR, RAPD-PCR, molecular detection, entomopathogenic fungi

## Abstract

In this study, molecular typing using Randomly Amplified Polymorphic DNA (RAPD-PCR) was conducted on 16 original isolates of *Metarhizium acridum* obtained from locusts (*Schistocerca piceifrons* ssp. *piceifrons*.) in Mexico (MX). The analysis included reference strains of the genus *Metarhizium* sourced from various geographical regions. The isolates were identified by phenotypic (macro and micromorphology) and genotypic methods (RAPD-PCR and Amplified Fragment Length Polymorphisms (AFLP), through a multidimensional analysis of principal coordinates (PCoA) and a minimum spanning network (MST). Subsequently, Sequences-Characterized Amplified Region (SCAR) markers were developed for the molecular detection of *M. acridum*, these markers were chosen from polymorphic patterns obtained with 14 primers via RAPD-PCR. Phenotypic and genotypic characterization identified the MX isolates as *M. acridum*. Of all the polymorphic patterns obtained, only OPA04 and OPA05 were chosen, which presented species-specific bands for *M. acridum*, and further utilized to create SCAR markers through cloning and sequencing of the specific bands. The specificity of these two markers was confirmed via Southern hybridization. The SCAR markers (Ma-160_OPA-05_ and Ma-151_OPA-04_) exhibit remarkable sensitivity, detecting down to less than 0.1 ng, as well as high specificity, as evidenced by their inability to cross-amplify or generate amplification with DNAs from other strains of *Metarhizium* (as *Metarhizium anisopliae*) or different genera of entomopathogenic fungi (*Cordyceps fumosorosea* and *Akanthomyces lecanii*). These SCAR markers yield readily detectable results, showcasing high reproducibility. They serve as a valuable tool, especially in field applications.

## 1. Introduction

The *Metarhizium* genus includes various species of fungi identified as entomopathogens since they can infect and parasitize a wide range of arthropods, which is why they are considered natural bioregulators for insect pests [1,2,3].

Likewise, due to the growing demand for sustainable agricultural practices, integrated pest management, and organic agriculture, among others, entomopathogens as microbial biocontrol agents have emerged as safe alternatives in terms of environmental impact and with low impact on human health, unlike chemical pesticides [4]. An example is the case of *Metarhizium acridum*, which has been successfully used to control acrids in Africa [5] and Australia [6]. In Mexico, since 2009, the application of this fungi to control *Schistocerca* spp. (nymphs and adults) has been applied in crops susceptible to this pest, such as sugar cane, corn, and fruit trees (https://www.gob.mx/senasica/documentos/hongos-entomopatogenos, accessed on 2 October 2023).

On the other hand, over the last decades, studies about the diversity, host affiliation, ecology, and distribution of species that comprised the genus *Metarhizium* have made substantial leaps forward. According to the last revision of *Metarhizum*, *M. acridum* belongs to the *M. anisopliae* complex, alongside *M. alvesii*, *M. anisopliae*, *M. baoshanense*, *M. brachyspermum*, *M. brittlebankisoides*, *M. brunneum*, *M. campsosterni*, *M. clavatum*, *M. globosum*, *M. guizhouense*, *M. humberi*, *M. indigotum*, *M. majus*, *M. lepidiotae*, *M. kalasinense*, *M. pingshaense*, *M. robertsii*, *M. phasmatodeae*, *M. gryllidicola*, and *M. sulphureum* [7].

As a result of continuous taxonomic changes and revisions using multilocus approaches, the identification of these entomopathogenic fungi used mainly in agricultural systems becomes crucial and complex, particularly to find the most efficient isolates in biocontrol programs, future commercial uses, such as products based on entomopathogenic fungi, or monitoring the presence of specific entomopathogenic fungi in soil, water, or other ecological niches [8].

To identify *Metarhizium*, several authors have used Restriction Fragment Length Polymorphism (RFLP) [9], Matrix-Assisted Laser Desorption/Ionization Time-Of-Flight (MALDITOF-MS) [10], and Loop-mediated Isothermal Amplification (LAMP) [11]. These methods have shown their effectiveness for a quick species validation. However, a rapid molecular identification tool that avoids sequencing can provide valuable information for various applications, particularly in identifying microorganisms in the field.

Several studies have demonstrated the usefulness of molecular tools in differentiating between particular species of the *Metarhizium* genus. Some of these studies could simultaneously track genetic differences within/among two or three species. For several years, Randomly Amplified Polymorphic DNA (RAPD) and Amplified Fragment Length Polymorphism (AFLP) have helped study the epidemiology of many pathogens [8,12,13,14,15] or the phylogenetic relationships between organisms [16]. The RAPD and AFLP methods have also been used to generate unique PCR products or amplicons in filamentous fungal species, or strains, of interest to be converted into species- or strain-specific Sequence-Characterized Amplified Region (SCAR) markers [17,18,19,20,21,22,23,24,25,26].

SCAR markers have also been used in the agricultural sector for the detection of phytopathogens, such as *Pseudocercospora eumusae*, the causal agent of eumusae leaf spot disease in bananas [27], *P. citricarpa*, the causal agent of citrus disease [28], and *Tilletia controversa* that causes wheat dwarf bunt [29].

Previous works show that these markers are a valuable tool in the identification and typing of microorganisms, so the objective of the present work was to develop SCAR markers based on the selection of a species-specific band for *M. acridum* from polymorphic patterns generated by RAPD-PCR obtained from monosporic cultures from Mexico and reference strains of different geographical origin.

## 2. Materials and Methods

### 2.1. Fungal Isolates

Sixteen original *M. acridum* isolates were obtained from the Collection of the Centro Nacional de Referencia de Control Biológico (CNRCB), Secretaría de Agricultura, Ganadería, Desarrollo Rural, Pesca y Alimentación (SAGARPA), Tecomán, Colima, Mexico (Table 1). All sixteen were isolated from locusts (*S. piceifrons* ssp. *piceifrons*) collected from pastures in Colima, including Socorro Island, Mexico (MX). Eleven reference strains of the genus *Metarhizium* from the Agricultural Research Service of Entomopathogenic Fungi, United States Department of Agriculture (ARSEF, USDA) and Commonwealth Scientific and Industrial Research Organization, Australia (CSIRO) were used as controls, representing the ten taxonomic clades proposed by Driver et al. [16] (Table 1). In addition, two isolates of *M. acridum* from Guatemala and one from MX, five isolates of *Cordyceps fumosorosea* and five of *Akanthomyces lecanii* were also used for SCAR specificity assays. Isolates were preserved in sterile water, mineral oil, and liquid nitrogen cryopreservation and then registered in the fungal collection of the Laboratorio de Micología Básica, Departamento de Microbiología y Parasitología, Facultad de Medicina, Universidad Nacional Autónoma de México, and registered at the World Federal Culture Collection (WFCC) as BMFM-UNAM 834. Isolates were cultured on potato dextrose agar (PDA, Bioxon, Mexico City, Mexico) slants and yeast peptone glucose broth (1% yeast extract, Yestal, Mexico City, Mexico; 2% peptone, Bioxon, Mexico City, Mexico; 4% glucose, Droguería Cosmopolita, Mexico City, Mexico) for further assays.

### 2.2. Phenotypic Characterization and Genotypic

Phenotypic characterization. Single-spore cultures were prepared from each isolate using the method described by Goettel and Inglis [30], modified as follows: 10 µL of a 104 conidia/mL suspension (100 conidia) was spread on a microscope slide containing 2.5 mL of 1.5% agarose. The slide was then examined under bright field illumination using a 10-X objective lens. When an isolated conidium was located, the slide was adjusted so that the target conidium was at the center of the field. The objective lens was then moved to one side and the microscope diaphragm was closed until only the target area was illuminated. A square of agarose (2 mm^2^) was marked around the illuminated point and the isolated conidium was again verified to be within the square using a 10- and 40-X objective lens. The agar square was excised and aseptically transferred onto PDA agar medium in a Petri dish. This procedure was repeated several times. The PDA Petri dish was observed daily under a stereoscopic microscope to confirm that the resulting culture was derived from a single spore. Ten monospore cultures were obtained from each isolate. Conidial size, colony growth rate, and spore production were measured for all monospore cultures and compared with those of the original isolate.

Genotypic characterization (RAPD and AFLP). RAPD-PCR tests were performed in a 25-µL reaction mixture. Ten nanograms of DNA were used with *Metarhizium* from Mexico, the monospore cultures, and reference strains. Each reaction contained target DNA, 1 X buffer, 2.5 mM MgCl_2,_ 200 µM concentration of each dNTP (Applied Biosystems Inc., Foster City, CA, USA), 200 pmol of each primer (Table 2) supplied by Qiagen^®^ (Hilden, Germany), and 1 U AmpliTaq Gold DNA polymerase (Applied Biosystems). Controls without DNA were also run with each set of reactions. PCR amplification was performed in an iCycler™ (Bio-Rad, Hercules, CA, USA), using the protocols previously described by Cobb and Clarkson (1993) [13]: one cycle for 5 min at 94 °C followed by 5 min at 94 °C, 2 min at 32 °C, and 2 min at 72 °C. This protocol was followed by 45 cycles of 1 min at 94 °C, 1 min at 35 °C, and 2 min at 72 °C to ensure full extension of all amplified products. After amplification, samples were maintained at 4 °C. All reactions were repeated twice to verify the reproducibility of all scored polymorphic bands. The products were subjected to 1.5% agarose gel electrophoresis and then stained with GelRed™ 10,000× Biotium (Fremont, CA, USA); DNA ladders of 100 and 1000 bp (molecular size marker) Invitrogen (Carlsbad, CA, USA) were used. Images of the gels were captured in a Synoptics Photodocumenter Syngene (Frederick, MD, USA).

All AFLP analyses in accordance with Vos et al. [31] were performed only with monospore cultures. DNA was restriction digested with the endonucleases EcoRI and MseI. After digestion, adaptors were ligated to the resulting fragments. The resulting fragments were preamplified with primers E (5′-GACTGCGTACCAATTC-3′) and M (5′-GACGATGAGTCCTGAGTAA-3′), after which a selective PCR was performed. The selective primers were identical to primer E or M, but they were extended with selective mono- and dinucleotides at the 3′- terminus. Three primer combinations were used: E+AC:M+A; E+C:M+A; and E+T:M+A. Primer E was radioactively labelled with ATP-P 32, and the amplified material was analyzed on 5% polyacrylamide slab gels.

### 2.3. Statistical Analysis

A matrix of presence (1) or absence (0) was constructed using the estimated molecular weights of the amplified RAPD-PCR and AFLP products, for the polyspore and monospore cultures, and another matrix for the MC and the 11 reference strains was also constructed. The genetic similarities among strains were calculated using the Jaccard coefficient [32]. The genetic diversity of polyspore and monosporic cultures was calculated using the Shannon index [33], considering each marker as a locus [34]. A phenogram based on each similarity matrix using the unweighted pair group method with arithmetic mean analysis (UPGMA) was generated. The cophenetic correlation coefficient (r) was calculated via the Mantel non-parametric test [35] to verify the reliability of the cluster analysis. To observe whether there were natural groups of *Metarhizium* isolates with clear geographical separation, multidimensional analyses of minimum spanning networks (MST) and principal coordinates (PCoA) were conducted based on the original similarity matrix. Multivariate statistical methods were applied using the NTSYS-PC program (Numerical taxonomy and multivariate analysis system) version 2.0 (Exeter Software, Setauket, NY, USA) [36].

### 2.4. Selection, Cloning, and Sequencing of RAPD Products

From the analysis of the polymorphic patterns obtained with the 14 primers used, species-specific bands were selected, which were excised from the agarose gel, and the DNA fragments were recovered using a QIAquick gel extraction kit (Qiagen), following the manufacturer’s protocol. An aliquot of the recovered DNA fragments was reamplified using the corresponding primer to verify that only a single band was excised. The bands were cloned into pGEM^®^-T Easy vector (Promega, Madison, WI, USA) and transformed into a competent *Escherichia coli* JM109. According to Sambrook et al. [37], plasmid DNA from recombinants was purified, followed by restriction digest with *Eco*R1 (Invitrogen, Carlsbad, CA, USA) to recover the insert DNA. The presence of the DNA fragment in the vector was checked using colony-PCR [38]. Plasmid DNA from positive clones was purified by the alkali lysis method, according to Sambrook et al. [37].

Hybridization experiments were performed to check the species-specificity of the cloned bands of the selected fungal markers. DNA obtained from isolates of *M. acridum* were transferred from agarose gels to Hybond-N+ nylon membranes (Amersham, UK) in a Vacuum Blotter 785 (Bio-Rad) and hybridized with the digoxigenin-labeled probe. Prehybridization and hybridization were carried out at 45 and 38 °C, respectively, followed by two final high-stringency washes at 45 °C, in accordance with Sambrook et al. [37]. The bands were visualized by a colorimetric method (NBT/BCIP stock solution; Roche, Basel, Switzerland) following the manufacturer’s protocol.

#### 2.4.1. Sequencing of Cloned Markers

The selected bands were sequenced at the Unidad de Biología Molecular, Instituto de Fisiología Celular, UNAM, using an ABI Prism 3100 automated DNA sequencer (Applied Biosystems). Sequence alignments were analyzed using the BLAST algorithm (BLAST+ 2.15.0) to check similarities among all fungal sequences deposited in the GenBank database [39].

#### 2.4.2. SCAR Primers Design

The sequences of the cloned markers were used to design pairs of SCAR primers using the Primer3 Input (http://frodo.wi.mit.edu/cgi-bin/primer3/primer3_www.cg, accessed on 20 October 2023). Details were followed carefully to avoid possible secondary structure or primer dimer generation and false priming and to match melting temperatures to achieve appropriate internal stability. The primers were synthesized using Sigma-Genosys (The Woodlands, TX, USA) (Figure 1).

### 2.5. PCR Conditions for SCAR Primers

SCAR-PCR mixtures (25 µL volume) for Ma-151_OPA-04_ marker assays consisted of 0.5 units of *AmpliTaq* Gold DNA Polymerase from Thermo Fisher Scientific Inc. (10 mM Tris–Cl, 50 mM KCl, pH 8.3, Waltham, MA, USA), 0.2 mM of each dNTP from Boehringer-Mannheim, 10 pmoles of each primer, 1.0 mM MgCl_2_, and 10 ng fungal DNA. Amplification was carried out under the following conditions: 94 °C for 3 min; 25 amplification cycles (1 min denaturation at 94 °C, 1 min annealing at 62 °C, 2 min of extension at 72 °C); and a final 5-min extension at 72 °C. PCR mixtures (25 µL volume) for the Ma-160_OPA-05_ marker assay consisted of 0.5 units of *AmpliTaq* Gold DNA Polymerase from Perkin-Elmer (10 mM Tris–HCl, 50 mM KCl, pH 8.3), 0.2 mM of each dNTP from Boehringer-Mannheim, 10 pmoles of each primer, 2.5 mM MgCl_2_, and 10 ng fungal DNA. Amplification was carried out under the following conditions: 94 °C for 3 min; 25 amplification cycles (1 min denaturation at 94 °C, 1 min annealing at 55 °C, 2 min of extension at 72 °C); and a final 5-min extension at 72 °C. Amplified products were resolved by electrophoresis in 1.5% agarose gels (Figure 1).

### 2.6. Sensitivity and Specificity of PCR Assays with the SCAR Marker

The sensitivity of the PCR system was determined by amplification from different amounts (from 5 ng to 1 pg) using *M. acridum* DNA as a template. Sensitivity assays were replicated at least twice. Sensitivity was defined as the lowest quantity of DNA template giving a product visible after agarose gel electrophoresis and UV transillumination.

The specificity of the SCAR markers was also tested by PCR of the genomic DNA from all *M. acridum* isolates used and the DNAs from 11 isolates of different species of *Metarhizium*, five isolates of *C. fumosorosea* and five isolates of *A. lecanii* (Figure 1).

## 3. Results

### 3.1. Phenotypic Characterization

All isolates and monospore cultures showed the typical colonial morphology and characteristic microscopic conidia of *M. acridum* (i.e., flat, white-colored colonies that gradually turned green, blue-green, or yellow-green over time, with branched conidiophores, clavate conidiogenous cells, and long chains of ellipsoidal basipetal conidia between 5.1 to 6.4 × 2.54 µm).

### 3.2. Genotypic Characterization

The combined analyses of RAPD and AFLP fragments showed a separate group from the 10 clades described by Driver et al. [16], and was formed by all isolates from MX. These isolates were strongly associated with strain FI-985 of *M. anisopliae* var. *acridum* (Australia) of clade 7 and with strain 1184 of *M*. *flavoviride* var. *flavoviride* (France) of clade 6. The similarity of the MX isolates with the reference strain of *M. acridum* analyzed by multidimensional analysis with PCoA and MST revealed the same separation between the isolates from MX and the reference strains of clades 1 through 10 as described by Driver et al. [16]. The PCoA and MST analysis shows a direct relationship between the MX isolates and reference strain FI985 of *M. anisopliae* var. *acridum* (Appendix A).

### 3.3. SCAR Marker in M. acridum

RAPD-PCR assays, using the 14 primers (Table 2), with MX isolates and reference strains, showed that the primers OPA-04 and OPA-05 yielded two easily detectable, well-resolved polymorphic bands of 526 and 239 bp (Ma526 and Ma239) and were always observed in *M. acridum* isolates from MX as well as the reference strains FI-985 and FI-987 (Figure 2A,B).

The polymorphic profiles were reproducible over repeated runs with sufficient intensity. These two species-specific RAPD markers were successfully cloned and sequenced. After verifying the transformation, two respective clones were chosen from each fragment.

From the RAPD-PCR obtained with the OPA-04 marker, the 526 bp band of the isolated MaPL5 was selected and from the RAPD-PCR obtained with the OPA-05 marker, the 289 bp band of the isolated MaPL-5 was selected. Specific bands were purified with a QIAquick gel extraction kit (Qiagen, Inc., Valencia, CA, USA), reamplified using primers OPA-04 and OPA-05 in the RAPD-PCR assay, and cloned into pGEM-T Easy Vector (Promega, Madison, WI, USA). The DNA insert harboring the plasmid with the expected molecular size was extracted, and this resulting insert (the SCAR marker) was used for Southern hybridization assays to confirm the presence of the selected marker in all isolates and reference strains from *M. acridum* (Appendix A); therefore, it can be inferred that the cloned fragments were derived from the amplified RAPD products of the genomic DNA of *M. acridum* isolates and not from contaminants of the same size.

### 3.4. Sequence Data Analysis

The sequences of the two Ma_526_ and Ma_289_ cloned fragments were analyzed using the BLAST algorithm (39) to check similarities among all fungal sequences deposited in the GenBank database. The results showed similarities with *Neurospora crassa*, *Saccharomyces cerevisiae*, *Fusarium tricinctum*, *Gibberella zeae*, and *Ophiostoma picea* (Appendix A). We eliminated the fungal homologous sequences found to attain the specificity of these SCAR marker pairs. Therefore, the 526 and 289 bp sequences were reduced to 153 and 239 bp, respectively. Based on the delimited sequences (153 and 239 bp) specific to *M. acridum*, two SCAR markers of 151 and 160 bp were designed, which were called Ma-151_OPA-04_ and Ma-160_OPA-05_ (Table 3). The sequences of these SCAR markers were submitted to GenBank (Accession numbers GU971357 and GU937436).

### 3.5. Sensitivity of SCAR Primers

The sensitivity test with both SCAR primers (Ma-151_OPA-04_-a and Ma-151_OPA-04_-b, and Ma-160_OPA-05_-a and Ma-160_OPA-05_-b) displayed the ability to detect trace amounts of DNA, amplifying the predicted amplicons of 151 and 160 bp from 5 ng to less than 0.1 ng of the genomic DNA of *M. acridum* (Figure 3).

### 3.6. Specificity of the SCAR Markers

SCAR primers Ma-151_OPA-04_-a and Ma-151_OPA-04_-b, and Ma-160_OPA-05_-a and Ma-160_OPA-05_-b amplified a unique fragment with the expected size of 151 and 160 bp, respectively, in all *M. acridum* isolates from MX and the two reference strains. No amplification products were observed when DNAs from reference strains of different *Metarhizium* species or other entomopathogenic fungi (five isolates of *C. fumosorosea* and five isolates of *A. lecanii*) were tested (Figure 4A,B).

To test the specificity of the oligonucleotides Ma-160_OPA-05_-a and Ma-160_OPA-05-b_, DNAs from *M. acridum* and five reference strains were used, showing that there was no non-specific amplification with the reference strains that do not belong to clade 7 (Figure 5). In the case of oligonucleotides Ma-151_OPA-04-a_ and Ma-151_OPA-04-b_, no non-specific amplification was also observed with the reference strains (Figure 5A,B).

## 4. Discussion

Within the genus *Metarhizium*, the species *M. acridum* (Driver & Milner), first identified by J.F. Bisch., S.A. Rehner and Humber, is one of the best-characterized species, and is known to naturally only infect orthopteran insects [16,40,41]. Like other species within this genus, *M. acridum* has a cosmopolitan distribution, mainly in tropical and subtropical regions [7]. Furthermore, *M. acridum* is more tolerant to physical stressors, such as ultraviolet light from solar radiation [42,43], than other *Metarhizium* species. *M. acridum* has been extensively studied for its potential as a biological control agent against locusts and grasshoppers, which can be significant agricultural pests [44]. This fungus has been researched and developed for its application in integrated pest management strategies, offering an environmentally friendly alternative to chemical insecticides [45].

As people broaden their perception of the environment and health, mycoinsecticides have become increasingly important substitutes for chemical insecticides due to their low toxicity, target specificity, and harmlessness to non-target organisms. As a microbial pesticide, *M. acridum* is widely used for the control of locusts and grasshoppers in Asia, Africa, and Australia, which relies on aerial conidia [5,6,46]. However, field studies on the efficacy and persistence of an introduced microbial control agent require assays to detect varying levels of the fungus in the field. The present study used the RAPD-PCR technique to obtain markers that differentiate *M. acridum* from its closely related species. These RAPD markers were converted to SCAR markers to develop a sensitive diagnostic assay for selective detection of *M. acridum* from diverse samples.

The isolates included in the study were identified phenotypically by macro- and micromorphology and genotypically, through the construction of a dendrogram using the polymorphic patterns obtained by RAPD and AFLP of the MX isolates and reference strains, which showed the relationship between the isolates of MX and the reference strains of *M. acridum*. This relationship was corroborated by multidimensional analysis with PCoA and MST. These methods demonstrated a high potential in distinguishing *Metarhizium* isolates.

Of the RAPD-PCR polymorphic patterns generated with the 14 primers, only those obtained with the OPA-04 and OPA-05 primers were selected, in which bands of 526 and 289 bp were evident, respectively; these were selected as candidates to obtain SCAR markers of *M. acridum*, which were studied by traditional methods of cloning, sequencing and sequence analysis. The first OPA-04(526)3 exhibited no more than 22 bp similarities when analyzed with BLAST against *G. zeae*, *N. crassa*, *S. cerevisiae*, and *F. tricinctum* sequences. On the other hand, the second sequence, OPA-05(289)5, showed similarities of 19 bp against *O. piceae*. In both cases, in the design of the specific oligonucleotides, sites shared with other fungal species were eliminated to increase specificity and reduce the possibilities of non-specific recognition (Appendix A).

Furthermore, the optimal amplification conditions were defined with the two pairs of specific oligonucleotides that generated the 160 and 151 bp fragments. The oligonucleotides designed from the OPA-05(289)5 sequence that generates the 160 bp band, when used to amplify DNA from isolates of *M. acridum* at a temperature of 55 °C, presented non-specificity with the reference strains 1914, FI-147, and FI-1029 belonging to the *M. majus*, *M. lepidiotum*, and *M. anisopliae*, respectively. However, when the temperature increased to 62 °C, the non-specificity of the reaction was eliminated. While, when the optimal conditions were established for the OPA-04(526)3 sequence with the specific oligonucleotides that generate the 151 bp band, when using an MgCl_2_ concentration of 2.5 mM, with DNA from isolates of *M. acridum* and reference strains, non-specific amplification was observed with strains 1914, 2037, FI-147, FI-1029, and FI-698 belonging to *M. majus*, *M. minus*, *M. lepidiotae*, *M. anisopliae*, and *M. novozealandicum*, respectively; however by decreasing the concentration of MgCl_2_ to 1 mM, the non-specific recognition was eliminated. The optimal determination of the reaction parameters allowed the specificity to be increased to 100%, with all isolates of *M. acridum*, including the reference strains FI-985 and FI-987, which correspond to *M. acridum* from AU (Australia) and NI (Nigeria), respectively.

On the other hand, the hybridization of the probes Ma-160_OPA-05_ and Ma-_151OPA-04_ on the transferred membranes with the RAPD-PCR polymorphic patterns generated from the oligonucleotides OPA-05 and OPA-04 showed the specificity themselves, since hybridization was only carried out with the isolates of *M. acridum.* Furthermore, it is essential to highlight that the oligonucleotide pairs are highly sensitive, since they detect less than 0.1 ng of DNA from *M. acridum*.

These SCAR markers can be instrumental in tracking the persistence and spread of *M. acridum* strains in the environment, particularly effective in Mexican agriculture since they were designed from autochthonous *M. acridum* isolates and did not amplify the DNA of unrelated isolates such as *C. fumosorosea* and *A. lecanii*, entomopathogenic fungi. Furthermore, they can be used for quality control in biopesticide production to ensure the purity and identity of strains used in commercial formulations [24].

Among the advantages of the SCAR markers is that compared to some high-throughput sequencing methods, they can be relatively cost-effective, making them accessible for laboratories with limited resources. Also, these markers give clear and easily detectable results and present a high reproducibility [47].

Regarding the limitations, each SCAR marker typically represents a single locus. For more complex genetic studies, other techniques that provide information on multiple loci simultaneously (e.g., microsatellites, SNP markers) may be more suitable. Further, these markers are limited to detecting variations in the known target region. They may not capture unknown genetic variations present elsewhere in the genome [48]. Despite this, SCAR markers remain valuable tools in molecular biology, especially for applications whose strengths align with the research objectives.

## 5. Conclusions

Two SCAR markers, Ma-160_OPA-05_ and Ma-151_OPA-04_, were obtained, amplifying a band from *M. acridum* of 160 and another 151 bp, respectively. In addition, they detect less than 0.1 ng of DNA from *M. acridum*. The designed oligonucleotides exhibit remarkable specificity, as evidenced by their inability to cross-amplify or generate amplification with DNAs from other strains of *Metarhizium* (as *M. anisopliae*) or different genera of entomopathogenic fungi (*C. fumosorosea* and *A. lecanii*).

## Figures and Tables

**Figure 1 jof-10-00269-f001:**
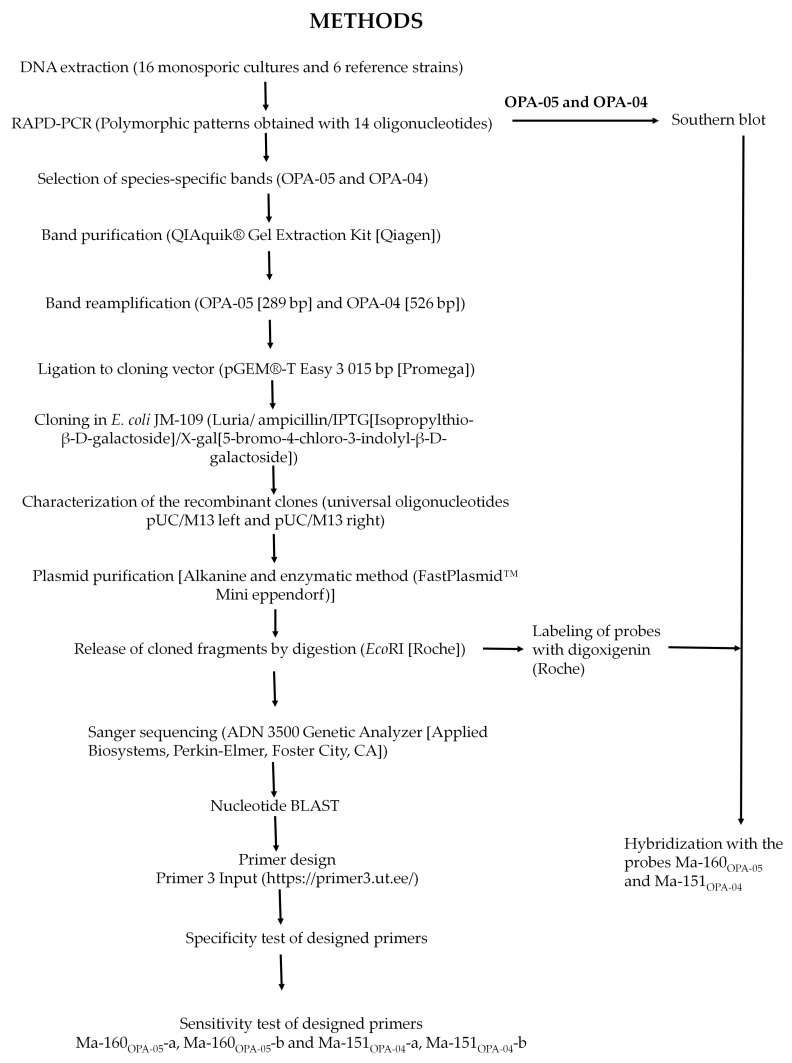
Flow diagram for obtaining SCAR markers.

**Figure 2 jof-10-00269-f002:**
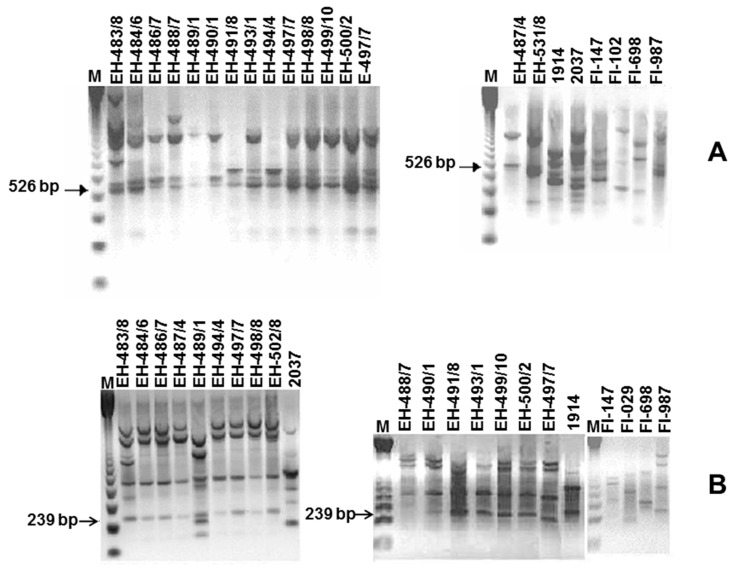
RAPD-PCR. The primers (**A**) OPA-04 and (**B**) OPA-05 yielded two easily detectable, well-resolved polymorphic bands (526 and 289 bp).

**Figure 3 jof-10-00269-f003:**
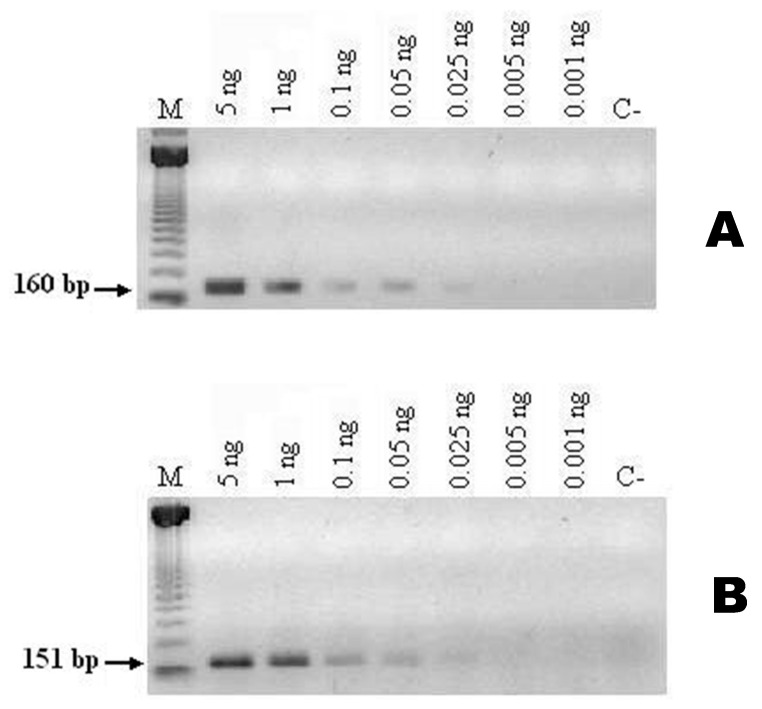
Sensitivity of the SCAR markers evaluated with different concentrations of genomic DNA of *M. acridum* isolate MaPL15 (EH-488/7). The markers were processed by PCR as described in Materials and Methods (**A**) with SCAR primers Ma-160_OPA-05_-a and Ma-160_OPA-05_-b and (**B**) with SCAR primers Ma-151_OPA-04_-a and Ma-151_OPA-04_-b. C−: negative control; M: 100 bp DNA ladder.

**Figure 4 jof-10-00269-f004:**
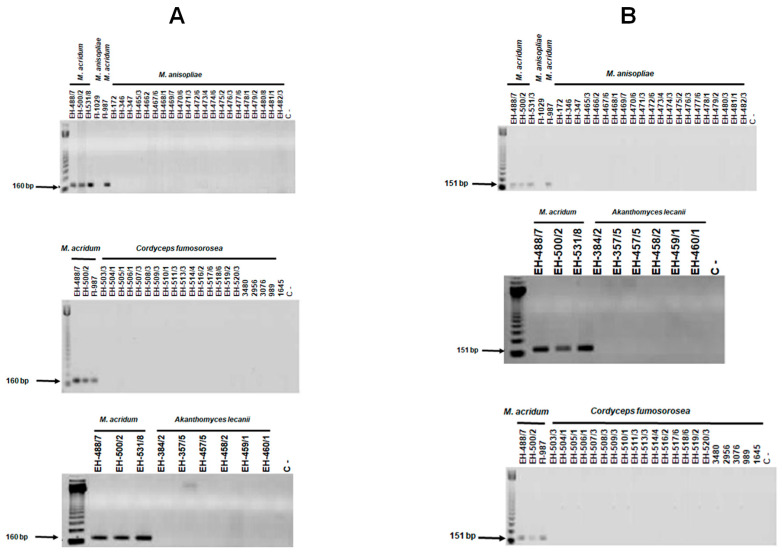
Specificity of the SCAR markers. PCR assays were performed with DNA from *M. acridum* and other entomopathogenic fungi (*M. anisopliae*, *C. fumosorosea*, and *A. lecanii*. The two SCAR markers were processed by PCR as described under Materials and Methods (**A**) with SCAR primers Ma-160_OPA-05_-a and Ma-160_OPA-05_-b and (**B**) with SCAR primers Ma-151_OPA-04_-a and Ma-151_OPA-04_-b. C−: negative control; M: 100 bp DNA ladder.

**Figure 5 jof-10-00269-f005:**
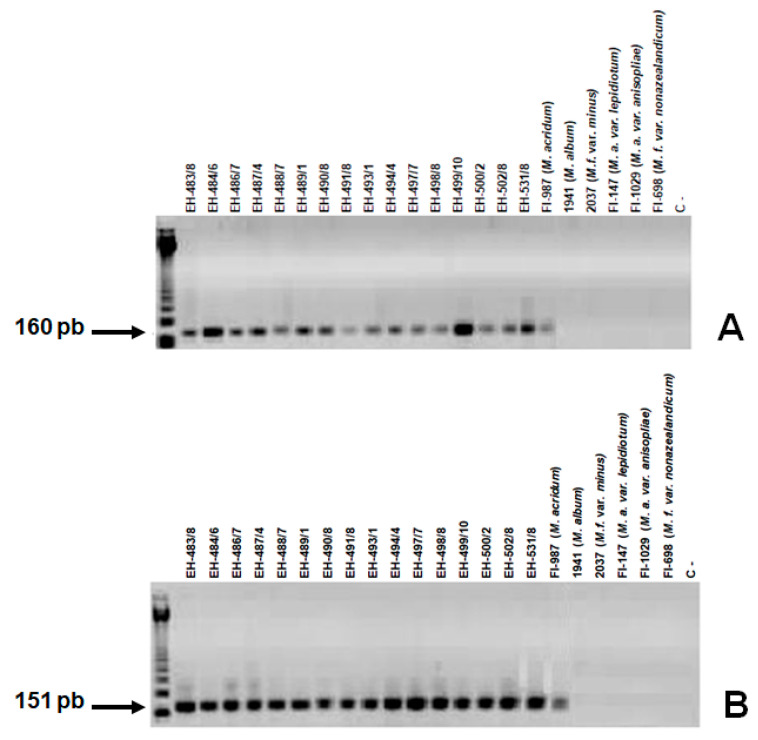
Specificity of the SCAR markers. PCR assays were performed with DNA from *M. acridum* and reference strains of clades 1, 3, 5, 8, and 9 (Driver et al. [16]) with the specific oligonucleotides (**A**) Ma-160_OPA-05-a_ and Ma-160_OPA-05-b_ and (**B**) with the specific oligonucleotides Ma-151_OPA-04_-a and Ma-151_OPA-04-b_. M: 123 bp DNA Ladder.

**Table 1 jof-10-00269-t001:** *Metarhizium* isolates and reference strains included in the study.

	Isolates		
CNRCB ^a^ Original Isolates Source Codes	UNAM ^b^ Monospore Cultures		
**México**			
MaPL5	EH-483/8		
MaPL8	EH-484/6		
MaPL13	EH-486/7		
MaPL14	EH-487/4		
MaPL15	EH-488/7		
MaPL16	EH-489/1		
MaPL18	EH-490/1		
MaPL20	EH-491/8		
MaPL22	EH-493/1		
MaPL26	EH-494/4		
MaPL29	EH-497/7		
MaPL31	EH-498/8		
MaPL37	EH-499/10		
MaPL38	EH-500/2		
MaPL40	EH-502/8		
MaPL32	EH-531/8		
**Reference Strains**			
**ARSEF ^c^** **CSIRO ^d^**	**Host**	**Country**	**Clade/Species**
1941 ^c^	*Nephotettix virescens* (Homoptera)	Philippines	1/*M. album*
2948 ^c^	Homoptera	Brasil	2/*M. flavoviride*
FI-698 ^d^	Lepidoptera	New Zealand	3/*M. novozealandicum*
FI-72 ^d^	*Pemphigus treherni* (Homoptera)	Britain	4/*M. pemphigi*
2037 ^c^	*Niliparvata lugens* (Homoptera)	Philippines	5/*M. minus*
1184 ^c^	*Otiorhynchus sulcatus* (Coleoptera)	France	6/*M. flavoviride*
FI-987 ^d^	*Otiorhynchus cavroisi* (Orthoptera)	Niger	7/*M. acridum*
FI-985 ^d^	*Austracris guttulosa*	Australia	7/*M. acridum*
FI-147 ^d^	*Lepidiota consobrina* (Coleoptera)	Australia	8/*M. lepidiotae*
FI-1029 ^d^	*S. gregária* (Orthoptera)	Eritrea	9/*M.* *anisopliae*
1914 ^c^	*Oryctes rhinoceros* (Coleoptera)	Philippines	10/*M. majus*

^a^ CNRCB = Centro Nacional de Referencia de Control Biológico, Mexico. ^b^ UNAM = Universidad Nacional Autónoma de México. ^c^ ARSEF = Agricultural Research Service of Entomopathogenic Fungi, United States Department of Agriculture, New York. ^d^ CSIRO = Commonwealth Scientific and Industrial Research Organization, Australia.

**Table 2 jof-10-00269-t002:** Primers used in RAPD-PCR assays.

Primer	Sequence	Reference
OPF06	5′-GGGAATTCGG-3′	Driver et al. [16]
OPF07	5′-CCGATATCCC-3′	
OPF08	5′-GGGATATCGG-3′	
OPF10	5′-GGAAGCTTGG-3′	
OPF01	5′-GGTCGGAGAA-3′	
OPF02	5′-TCGGACGTGA-3′	
OPA01	5′-CAGGCCCTTC-3′	Cobb and Clarkson [13]
OPA04	5′-AATCGGGCTG-3′	
OPA05	5′-AGGGGTCTTG-3′	
OPA08	5′-GTGACGTAGG-3′	
OPA09	5′-GGGTAACGCC-3′	
OPA10	5′-GTGATCGCAG-3′	
OPA16	5′-AGCCAGCGAA-3′	
OPA19	5′-CAAACGTCGG-3′	

**Table 3 jof-10-00269-t003:** Sequences of SCAR primers derived from RAPD markers diagnostic for *M. acridum*.

SCAR	Primers	Sequence (5′-3′)
Ma-151_OPA-04_	Ma-151_OPA-04_-a	TGGTCAGAGCTCACGTCCAC
	Ma-151_OPA-04_-b	TGAAGACATTCAGAGGCCAGT
Ma-160_OPA-05_	Ma-160_OPA-05_-a	TGCGCCTAGGATGCTTGTTA
	Ma-160_OPA-05_-b	GGCGACGCTCATATTCAACT

## Data Availability

Data are contained within the article and Appendix A.

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
