# Peer review of "Development of SCAR Markers for Genetic Authentication of Metarhizium acridum"

_jof, 2024, doi:10.3390/jof10040269_

Round 1

Reviewer 1 Report

Dear authors! Thank you for interesting work “Development of SCAR markers for genetic authentication of Metarhizium acridum.

The work is done at a good level. The scheme of the experiment is correct and does not raise any questions.

The main remark to the work:

Insufficient characterization of the sequences obtained for SCAR markers.

The authors mention that certain sequences (accession numbers GU971357 and GU937436) at 526 and 289 bp contain substitutions found during BLAST with other fungi (Gibberella zeae, Neurospora crassa, Saccharomyces cerevisiae, and Fusarium tricinctum). However, they do not cite these alignments in the article or ESM. Also, the authors do not describe whether these sequences are coding or non-coding. There was not ORF searching for these sequences.

This is an important point to understand the potential rate of mutation accumulation in these sequences. For example, in non-coding regions the rate of mutation accumulation (point substitutions) can be quite high. And the SCAR primers selected by authors will be not working after a short period.

The section 3.4 "PCR conditions for SCAR primers" should be referred to the section 2."Materials and methods."

Lines 299-325 should be moved from the “Discussion” section to the “Results” section or removed at all.

Should be the “Gibberella zeae” instead of “Gibberella zea”.

Author Response

Reviewer 1

Comments for Authors

Advice for completing your review can be found at: https://www.mdpi.com/reviewers#Review_Report

Major comments

Dear authors! Thank you for interesting work “Development of SCAR markers for genetic authentication of Metarhizium acridum.

The work is done at a good level. The scheme of the experiment is correct and does not raise any questions.

The main remark to the work:

Insufficient characterization of the sequences obtained for SCAR markers.

The authors mention that certain sequences (accession numbers GU971357 and GU937436) at 526 and 289 bp contain substitutions found during BLAST with other fungi (Gibberella zeaeNeurospora crassaSaccharomyces cerevisiae, and Fusarium tricinctum). However, they do not cite these alignments in the article or ESM. Also, the authors do not describe whether these sequences are coding or non-coding. There was not ORF searching for these sequences.

This is an important point to understand the potential rate of mutation accumulation in these sequences. For example, in non-coding regions the rate of mutation accumulation (point substitutions) can be quite high. And the SCAR primers selected by authors will be not working after a short period.

Answer: The sequences were analyzed with the ORF Finder program and both sequences show coding regions, the GU971357 sequence has 6 ORFs, while the GU937436 sequence has 8 ORFs. This information was added to the manuscript. Sequence alignments were added in Supplementary Material.

Detail comments

The section 3.4 "PCR conditions for SCAR primers" should be referred to the section 2."Materials and methods."

Lines 299-325 should be moved from the “Discussion” section to the “Results” section or removed at all.

Answer: The text was transferred to the materials and methods section.

Should be the “Gibberella zeae” instead of “Gibberella zea”.

Answer: It was corrected in the text.

Reviewer 2 Report

The authors investigate the possibilities of molecular diagnostics specific for the entomopathogenic fungal species Metarhizium acridum and present two RAPD-derived SCAR primer pairs amplifying, respectively, a 160bp and 151bp PCR product specifically from genomic DNA samples of strains/isolates belonging to M. acridum, and not from those belonging to further Metarhizium species or other entomopathogenic and/or taxonomically related fungal genera.

The manuscript is reasonably well structured and clearly written, the introduction section including citations is appropriate as are those parts of the discussion section dedicated to the application context of Metarhizium in biocontrol and the use of a species-specific diagnostics tool in this context, methods are generally well described, and technically experiments appear well performed.

Unfortunately, the same is not true for the results section that contains several severe inconsistencies that might in part be due to suboptimal data presentation but will be connected to an inappropriate design of parts of the study.

Comparatively minor issues:

“Base pairs” is abbreviated “bp”, not “pb”, please check.

Sensitivity assay for SCAR primer pairs, both in methods and results (Legend to Figure 3): the manuscript text states three times that the assay was performed with “DNA of M. acridum”. Please indicate the strain or isolate used.

Conclusion that M. acridum is detected from up to 25pg of DNA appears very optimistic interpretation not really supported by gels shown in Figure 3. The band is so weak that it would not be judged reliable for diagnostics. Robust detection appears to have a threshold between 1ng and 0.1ng according to gels presented. This represents a valuable result, too.

Lines 134-136 state that different amounts of DNA were used for different strains in RAPD-PCR assays. Why this?

Lines 213-217 might be deleted as i)the naming of clones is rather obvious for a reader following the paper, and ii) later no critical reference is made to these designations.

Legends to Figures 3 and 4: “SCAR” primer, not “SAR” primer. Text “A) with SCAR primers” is duplicated, please delete.

Lines 134-136: RAPD-PCR used 10ng of extracted DNA with some strains, only 1ng with others. This si a difference that might determine the outcome. Authors should give a hint why they did so.

Figure 1 might well be presented as Suppl Figure.

Table 2 might be presented as Suppl Table.

DNA giving rise to 2 polymorphic RAPD bands identified as presumably species-specific was cloned to an E. coli vector and used for sequencing and hybridization probe generation. The corresponding explanations (lines 153ff and 209ff) do not reveal from which fungal strain or strains these probes and sequences were obtained and used for hybridization. This should be clearly stated.

If sequences were obtained from several fungal strains, sequence similarity data should be presented and discussed.

Potentially major flaws in data presentation and/or design of the study

Strains used in the study:

The authors use 16 own Mexican M. acridum isolates plus a set of 11 reference strains from culture collections, comprising two further M. acridum strains (FI-985 and FI-987, from different geographic origins). If I am not mistaken FI-987 is the nomenclatural type strain of the species M. acridum; if correct, this should be stated. This strain selection appears generally reasonable: as in their interpretation the authors stress the species-specificity of their SCAR approach, it appears appropriate to compare M. acridum strains mainly to other Metarhizium species.

However, the authors do not present any conclusive evidence that the 16 Mexican isolates belong to M. acridum. A brief discussion of the “typical colonial morphology and characteristic microscopic conidia of M. acridum” found in the monospore cultures derived from the isolates (lines 195-199) is hardly sufficient. It is widely accepted that phenoptypic characteristics alone are not sufficient for species level assignment in Metarhizium. Recognized marker gene sequences (as ef1a, RPB1, RPB2, 5TEF) for Metarhizium molecular taxonomy should be compared to reference sequences including those of the M. acridum type strain. If respective data for the Mexican isolates have been published elsewhere the source should be duly cited. If not, the authors should add a respective molecular taxonomy based assignment (for instance an ef1a-rpb1-rpb2 concatenation or a 5TEF tree) to the manuscript.

RAPD-PCR and polymorphic band selection:

Figure 2 displays RAPD patterns for the 16 Mexican isolates plus 6 out of 11 Metarhizium reference strains. In particular, the pattern of M. acridum reference FI-985 is not presented. However, the text of the results section (lines 203-205) states that the two bands in question (Ma526 and Ma239) “were always observed in M. acridum isolates from MX as well as the reference strains FI-985 and FI-987” and refers for support of the claim to “Figure 2 A and B”. Moreover, as the intended species-specificity of the band does not only require that it produced in all M. acridum strains investigated but is at the same time absent from all other reference strains, it would appear logical that RAPD results are displayed for all Metarhizium reference strains listed in Table 1. As “polymorphic profiles were reproducible over repeated runs” and therefore RAPD assays were repeated, there should have been occasion to get suitable gel pictures. If authors did not assay all of the systematically chosen reference strains “representing the ten taxonomic clades proposed by Driver et al. 2009” (lines 94-95) in RAPD-PCR, they should urgently explain why they did so.

Southern hybridization:

Authors hybridized the cloned SCAR marker DNA against RAPD products of the “18 M. acridum isolates”, i.e. the 16 isolates from Mexico plus both M. acridum references. Of course this setting is not suitable to assess species-specificity: for this purpose, RAPD products from non – M. acridum references should be included in the hybridization. However, the discussion section states  that hybridization experiments "showed the specificity … since hybridization was only carried out with the isolates of M. acridum”. This statement appears unfounded by the experimental settings, specificity of the probe was not assessed.

Data of Southern hybridization experiments might be shown.

SCAR marker specificity:

Figure 4 presents PCR results with both diagnostic SCAR primer pairs designed by the authors. Choice of fungal strains used for this experiment looks rather confusing and requires explanations not given in the dry 6 lines the results section dedicates to these complex experimental data.

It is the manuscript’s central claim that both sets of SCAR primers amplified a unique fragment of expected apparent size “in all (18) M. acridum isolates” whereas “no amplification was observed … from different Metarhizium species or other entomopathogenic fungi”. However, on the one hand, positive amplification results for both SCAR markers are shown for only 3 Mexican isolates and reference strain FI-987. If amplification worked from all M. acridum listed in Table 1, then authors should please present the data. Their specificity claim depends on this result.

On the other hand, negative amplification results are – as far as I understand labelling of Figure 4 – presented for a single out of 9 non – M. acridum reference strains listed in Table 1: M. anisopliae FI-1029 (labelled – into the bargain - as M. acridum in Figure 4, seemingly contradicting the central claim cited above). Why were the further reference strains representing a wider spectrum of Metarhizium species not assayed or data not presented?

Moreover, Figure 4 appears to comprise the negative (!) diagnostic PCR result for three further Mexican strains addressed as M. acridum (!) that are – as far as I see – not mentioned earlier or later in the study. Authors should urgently clarify this not only apparent confusion.

Furthermore, Figure 4 presents negative results for a comparatively large number of – obviously - Mexican isolates not described in materials and methods as representing M. anisopliae (18), C. fumosorosea (16) and A. lecanii (5). On which basis have these isolates been assigned to these species and chosen as reference for SCAR-PCR specificity assays? What is the particular interest of Cordyceps fumosorosea and Akanthomyces lecanii for the purpose of this paper?

In conclusion, the authors’ strategy to firstly systematically select a set of reference strains for critical method evaluation, and then omit and step by step substitute chosen reference strains by other strains and own isolates puts in doubt the consistency of the study and the central claim of the manuscript.

Author Response

Reviewer 2

Comments for Authors

Advice for completing your review can be found at: https://www.mdpi.com/reviewers#Review_Report

Major comments

The authors investigate the possibilities of molecular diagnostics specific for the entomopathogenic fungal species Metarhizium acridum and present two RAPD-derived SCAR primer pairs amplifying, respectively, a 160bp and 151bp PCR product specifically from genomic DNA samples of strains/isolates belonging to M. acridum, and not from those belonging to further Metarhizium species or other entomopathogenic and/or taxonomically related fungal genera.

El manuscrito está razonablemente bien estructurado y claramente escrito, la sección de introducción que incluye citas es apropiada al igual que las partes de la sección de discusión dedicadas al contexto de aplicación de Metarhizium en biocontrol y el uso de una herramienta de diagnóstico específica de especie en este contexto, los métodos son En general, están bien descritos y técnicamente los experimentos parecen bien realizados.

Desafortunadamente, no ocurre lo mismo con la sección de resultados, que contiene varias inconsistencias graves que podrían deberse en parte a una presentación subóptima de los datos, pero que estarán relacionadas con un diseño inadecuado de partes del estudio.

Detail comments

Comparatively minor issues:

“Base pairs” is abbreviated “bp”, not “pb”, please check.

Answer: It was corrected in the text.

Sensitivity assay for SCAR primer pairs, both in methods and results (Legend to Figure 3): the manuscript text states three times that the assay was performed with “DNA of M. acridum”. Please indicate the strain or isolate used.

Answer: The information was added in Figure 2 (the isolate used was MaPL15 (EH-488/7).

Conclusion that M. acridum is detected from up to 25pg of DNA appears very optimistic interpretation not really supported by gels shown in Figure 3. The band is so weak that it would not be judged reliable for diagnostics. Robust detection appears to have a threshold between 1ng and 0.1ng according to gels presented. This represents a valuable result, too.

Answer: Based on your observation, we have decided to consider that the amount of DNA detected by the primers is less than 0.1 ng, since below this amount the amplification is little observed, that is, it continues to amplify. This was changed in the text.

Lines 134-136 state that different amounts of DNA were used for different strains in RAPD-PCR assays. Why this?

Answer: Sorry, it was a mistake, 10 ng of DNA from all reference strains was used. The phrase “1914 and FI-147, whereas 1 ng of DNA was used with the remaining reference strains” in the manuscript was removed.

Lines 213-217 might be deleted as i) the naming of clones is rather obvious for a reader following the paper, and ii) later no critical reference is made to these designations.

Answer: This paragraph was removed.

Legends to Figures 3 and 4: “SCAR” primer, not “SAR” primer. Text “A) with SCAR primers” is duplicated, please delete.

Answer: The error in the figure legends was corrected.

Lines 134-136: RAPD-PCR used 10ng of extracted DNA with some strains, only 1ng with others. This si a difference that might determine the outcome. Authors should give a hint why they did so.

Answer: As mentioned before, it was a mistake. 10 ng of DNA was used in the RAPD assays since if different amounts are used, it could influence the polymorphic patterns generated.

Figure 1 might well be presented as Suppl Figure.

Answer: Figure 1 was not added as supplementary material since we consider that it will be a useful guide for readers.

DNA giving rise to 2 polymorphic RAPD bands identified as presumably species-specific was cloned to an E. coli vector and used for sequencing and hybridization probe generation. The corresponding explanations (lines 153ff and 209ff) do not reveal from which fungal strain or strains these probes and sequences were obtained and used for hybridization. This should be clearly stated.

If sequences were obtained from several fungal strains, sequence similarity data should be presented and discussed.

Answer: This information is specified in the Results section: “RAPD-PCR assays, using the 14 primers (Table 2), with MX isolates and reference strains, showed that the primers OPA-04 and OPA-05 yielded two easily detectable, well-resolved polymorphic bands of 526 and 239 bp (Ma526 and Ma239) and were always observed in M. acridum isolates from MX as well as the reference strains FI-985 and FI-987”.

Potentially major flaws in data presentation and/or design of the study Strains used in the study:

The authors use 16 own Mexican M. acridum isolates plus a set of 11 reference strains from culture collections, comprising two further M. acridum strains (FI-985 and FI-987, from different geographic origins). If I am not mistaken FI-987 is the nomenclatural type strain of the species M. acridum; if correct, this should be stated. This strain selection appears generally reasonable: as in their interpretation the authors stress the species-specificity of their SCAR approach, it appears appropriate to compare M. acridum strains mainly to other Metarhizium species.

Answer: In Table S1, it is indicated that strains FI-985 and FI-987 are M. acridum and correspond to Clade 7 of the classification of Driver et al (2000). Table S1 was corrected.

However, the authors do not present any conclusive evidence that the 16 Mexican isolates belong to M. acridum. A brief discussion of the “typical colonial morphology and characteristic microscopic conidia of M. acridum” found in the monospore cultures derived from the isolates (lines 195-199) is hardly sufficient. It is widely accepted that phenoptypic characteristics alone are not sufficient for species level assignment in Metarhizium. Recognized marker gene sequences (as ef1a, RPB1, RPB2, 5TEF) for Metarhizium molecular taxonomy should be compared to reference sequences including those of the M. acridum type strain. If respective data for the Mexican isolates have been published elsewhere the source should be duly cited. If not, the authors should add a respective molecular taxonomy based assignment (for instance an ef1a-rpb1-rpb2 concatenation or a 5TEF tree) to the manuscript.

Answer: We agree with you that phenotypic characterization is not enough to identify the species, so in this work, all isolates, in addition to being phenotypically characterized through macro and micromorphology, were genotypically characterized using RAPD-PCR and AFLP methods. The results of the genotypic characterization were included in the manuscript.

RAPD-PCR and polymorphic band selection:

Figure 2 displays RAPD patterns for the 16 Mexican isolates plus 6 out of 11 Metarhizium reference strains. In particular, the pattern of M. acridum reference FI-985 is not presented. However, the text of the results section (lines 203-205) states that the two bands in question (Ma526 and Ma239) “were always observed in M. acridum isolates from MX as well as the reference strains FI-985 and FI-987” and refers for support of the claim to “Figure 2 A and B”. Moreover, as the intended species-specificity of the band does not only require that it produced in all M. acridum strains investigated but is at the same time absent from all other reference strains, it would appear logical that RAPD results are displayed for all Metarhizium reference strains listed in Table 1. As “polymorphic profiles were reproducible over repeated runs” and therefore RAPD assays were repeated, there should have been occasion to get suitable gel pictures. If authors did not assay all of the systematically chosen reference strains “representing the ten taxonomic clades proposed by Driver et al. 2009” (lines 94-95) in RAPD-PCR, they should urgently explain why they did so.

Southern hybridization:

Authors hybridized the cloned SCAR marker DNA against RAPD products of the “18 M. acridum isolates”, i.e. the 16 isolates from Mexico plus both M. acridum references. Of course this setting is not suitable to assess species-specificity: for this purpose, RAPD products from non – M. acridum references should be included in the hybridization. However, the discussion section states  that hybridization experiments "showed the specificity … since hybridization was only carried out with the isolates of M. acridum”. This statement appears unfounded by the experimental settings, specificity of the probe was not assessed.

Data of Southern hybridization experiments might be shown.

Answer: The results of the hybridization assays were added to the supplementary material.

SCAR marker specificity:

Figure 4 presents PCR results with both diagnostic SCAR primer pairs designed by the authors. Choice of fungal strains used for this experiment looks rather confusing and requires explanations not given in the dry 6 lines the results section dedicates to these complex experimental data.

It is the manuscript’s central claim that both sets of SCAR primers amplified a unique fragment of expected apparent size “in all (18) M. acridum isolates” whereas “no amplification was observed … from different Metarhizium species or other entomopathogenic fungi”. However, on the one hand, positive amplification results for both SCAR markers are shown for only 3 Mexican isolates and reference strain FI-987. If amplification worked from all M. acridum listed in Table 1, then authors should please present the data. Their specificity claim depends on this result.

Answer: Results from specificity assays with other reference strains were included in the manuscript.

On the other hand, negative amplification results are – as far as I understand labelling of Figure 4 – presented for a single out of 9 non – M. acridum reference strains listed in Table 1: M. anisopliae FI-1029 (labelled – into the bargain - as M. acridum in Figure 4, seemingly contradicting the central claim cited above).

Answer: We appreciate your observation, and you are indeed right, there was an error in the figure, since strain FI-1029 is not M. acridum, it did not amplify, and as shown in supplementary table S1 it corresponds to the species M anisoplae. This figure was corrected.

Why were the further reference strains representing a wider spectrum of Metarhizium species not assayed or data not presented?

Answer: To test the specificity of the oligonucleotides Ma-160OPA-05-a and Ma-160OPA-05-b, DNAs from M. anisopliae var. acridum and reference strains, showing that there is no nonspecific amplification with any of the DNAs of the reference strains that do not belong to var. acridum This information was added to the results.

Moreover, Figure 4 appears to comprise the negative (!) diagnostic PCR result for three further Mexican strains addressed as M. acridum (!) that are – as far as I see – not mentioned earlier or later in the study. Authors should urgently clarify this not only apparent confusion.

Furthermore, Figure 4 presents negative results for a comparatively large number of – obviously - Mexican isolates not described in materials and methods as representing M. anisopliae (18), C. fumosorosea (16) and A. lecanii (5). On which basis have these isolates been assigned to these species and chosen as reference for SCAR-PCR specificity assays? What is the particular interest of Cordyceps fumosorosea and Akanthomyces lecanii for the purpose of this paper?

Answer: Information was added to the “fungal isolates” section in materials and methods about isolates not included in table 1.

On the other hand, the isolates of Cordyceps fumosorosea and Akanthomyces lecanii were used, because they can be found in the environment and can give a cross-reaction with the markers obtained.

In conclusion, the authors’ strategy to firstly systematically select a set of reference strains for critical method evaluation, and then omit and step by step substitute chosen reference strains by other strains and own isolates puts in doubt the consistency of the study and the central claim of the manuscript.

Answer: We appreciate your valuable comments and regret the errors made to describe the results clearly, however, we hope that in this new version, the way to present the results will be clearer.

Round 2

Reviewer 2 Report

The authors have responded with mostly appropriate and in part substantial changes to most of the issues, doubts and suggestions raised in the first round of reviewing and have considerably improved the manuscript.

However, the following suggestion was not appropriately addressed by the authors, probably due to a mutual misunderstanding:

Original Reviewer Comment: “DNA giving rise to 2 polymorphic RAPD bands identified as presumably species-specific was cloned to an E. coli vector and used for sequencing and hybridization probe generation. The corresponding explanations (lines 153ff and 209ff of the original manuscipt) do not reveal from which fungal strain or strains these probes and sequences were obtained and used for hybridization. This should be clearly stated. If sequences were obtained from several fungal strains, sequence similarity data should be presented and discussed.”

Authors Reply: “This information is specified in the Results section: “RAPD-PCR assays, using the 14 primers (Table 2), with MX isolates and reference strains, showed that the primers OPA-04 and OPA-05 yielded two easily detectable, well-resolved polymorphic bands of 526 and 239 bp (Ma526 and Ma239) and were always observed in M. acridum isolates from MX as well as the reference strains FI-985 and FI-987”.”

My question was NOT with which strains or isolates the RAPD assay produced the species-specific bands, but from which M. acridum strain or strains these were cloned, sequenced and further developed into a SCAR marker and hybridization probe. It appears clear from subsequent experiments that only one probe Ma-160OPA-05 and only one probe Ma-151OPA-04  were used. So it should be indicated from which strain has this sample been derived – or, if the sequence has been determined from several strains, sequence comparison data should be revealed (for instance stating from which strains sequences were determined and that they were all identical or XX% similar).

--------------------------

A second issue that remains is the lack of conclusive evidence that the 16 Mexican isolates belong to M. acridum. The authors newly present in this modified version of the manuscript a MST and PcoA approach based on combined RAPD and AFLP pattern analyses. This analysis is in itself technically well done. However, I have some doubt that this type of analysis is – under the settings chosen or better imposed by the study design (for instance the references included etc.) – in principle suitable to assess the species level assignment of the Mexican isolates. Moreover, the results obtained clearly demonstrate (and the authors state it in lines 240-248) that i) Mexican isolates from a tight group and ii) this group is equally distant from both M. acridum and M. flavoviride references. As M. flavoviride is rather distantly related with M. acridum, this is far from a clear-cut result.

I highly appreciate the bulk of work the authors have invested in this part of the analysis, but nevertheless would like to invite them to consider if determination of, for instance, the 5TEF sequences of at least several of the Mexican isolates and comparison with 5TEF references abundantly present in the GenBank database might not be a feasible step to put the entire papers valuable message on undoubtedly solid ground.

------------------------------------

Suggestions/comments of minor importance:

“Characterization Phenotypic / Genotypic” should be inversed: “Phenotypic / Genotypic Characterization”

Legend to Suppl Figure S1: “Supplementary Table 1” should probably read “Table 1”. Otherwise Suppl. Table S1 is lacking.

Suppl. Figures S4 and S5 might be indicated in the paragraph lines 272-281.

It is not really clear from the main text which hybridization probe has been used for Southerns displayed in Supplementary Figures S2 and S3, the whole cloned RAPD bands or the SCAR marker developed out of it. If blots were hybridized with the SCAR marker (as is indicated in the respective Figure Legends), paragraph 263-269 (presenting hybridzation results) might better be reorganized after paragraph 272-281 and Table 3 (presenting SCAR marker development).

Author Response

Major comments

The authors have responded with mostly appropriate and in part substantial changes to most of the issues, doubts and suggestions raised in the first round of reviewing and have considerably improved the manuscript.

Detail comments

However, the following suggestion was not appropriately addressed by the authors, probably due to a mutual misunderstanding:

Original Reviewer Comment: “DNA giving rise to 2 polymorphic RAPD bands identified as presumably species-specific was cloned to an E. coli vector and used for sequencing and hybridization probe generation. The corresponding explanations (lines 153ff and 209ff of the original manuscipt) do not reveal from which fungal strain or strains these probes and sequences were obtained and used for hybridization. This should be clearly stated. If sequences were obtained from several fungal strains, sequence similarity data should be presented and discussed.”

Authors Reply: “This information is specified in the Results section: “RAPD-PCR assays, using the 14 primers (Table 2), with MX isolates and reference strains, showed that the primers OPA-04 and OPA-05 yielded two easily detectable, well-resolved polymorphic bands of 526 and 239 bp (Ma526 and Ma239) and were always observed in M. acridum isolates from MX as well as the reference strains FI-985 and FI-987”.”

My question was NOT with which strains or isolates the RAPD assay produced the species-specific bands, but from which M. acridum strain or strains these were cloned, sequenced and further developed into a SCAR marker and hybridization probe. It appears clear from subsequent experiments that only one probe Ma-160OPA-05 and only one probe Ma-151OPA-04  were used. So it should be indicated from which strain has this sample been derived – or, if the sequence has been determined from several strains, sequence comparison data should be revealed (for instance stating from which strains sequences were determined and that they were all identical or XX% similar).

Answer: This information was added to the Results section.

A second issue that remains is the lack of conclusive evidence that the 16 Mexican isolates belong to M. acridum. The authors newly present in this modified version of the manuscript a MST and PcoA approach based on combined RAPD and AFLP pattern analyses. This analysis is in itself technically well done. However, I have some doubt that this type of analysis is – under the settings chosen or better imposed by the study design (for instance the references included etc.) – in principle suitable to assess the species level assignment of the Mexican isolates. Moreover, the results obtained clearly demonstrate (and the authors state it in lines 240-248) that i) Mexican isolates from a tight group and ii) this group is equally distant from both M. acridum and M. flavoviride references. As M. flavoviride is rather distantly related with M. acridum, this is far from a clear-cut result.

I highly appreciate the bulk of work the authors have invested in this part of the analysis, but nevertheless would like to invite them to consider if determination of, for instance, the 5TEF sequences of at least several of the Mexican isolates and comparison with 5TEF references abundantly present in the GenBank database might not be a feasible step to put the entire papers valuable message on undoubtedly solid ground.

Answer: We agree that there may be a reasonable doubt regarding whether the isolates in this study belong to the species M. acridum, however, currently, there is no single method (morphological, physiological, or molecular) that works perfectly in species recognition, so it is considered that the more parameters that are included in the identification process, the more solid the species classification will be. So the phenotypic identification combined with the characterization of the isolates through RAPD using 14 oligonucleotides, in addition to AFLP with three combinations of selective oligonucleotides, seems to us to provide sufficient information, if not ideal, to affirm that the isolates belong to M. acridum, since these methods have been useful to evaluate phylogenetic relationships among other organisms as well.

On the other hand, the results of the joint analysis of the polymorphic patterns obtained with both RAPD and AFLP showed that the MX isolates formed a separate group from the reference strains and showed a similarity of 73% with the M. anisopliae strain and a 99% similarity to M. flavoviride. However, when the minimum span tree analysis was performed, it indicated that MX isolates had a more direct relationship with M. acridum than with M. flavoviride, which reinforces the usefulness of these methods to discriminate between different species.

Suggestions/comments of minor importance:

 “Characterization Phenotypic / Genotypic” should be inversed: “Phenotypic / Genotypic Characterization”

Answer: It was corrected in the text.

Legend to Suppl Figure S1: “Supplementary Table 1” should probably read “Table 1”. Otherwise Suppl. Table S1 is lacking.

Answer: It was corrected in the legend of Supplementary Figure1.

Suppl. Figures S4 and S5 might be indicated in the paragraph lines 272-281.

Answer: The information was added in the text.

It is not really clear from the main text which hybridization probe has been used for Southerns displayed in Supplementary Figures S2 and S3, the whole cloned RAPD bands or the SCAR marker developed out of it. If blots were hybridized with the SCAR marker (as is indicated in the respective Figure Legends), paragraph 263-269 (presenting hybridzation results) might better be reorganized after paragraph 272-281 and Table 3 (presenting SCAR marker development).